# Proton Affinity in the Chemistry of Beta-Octamolybdate: HPLC-ICP-AES, NMR and Structural Studies

**DOI:** 10.3390/molecules27238368

**Published:** 2022-11-30

**Authors:** Victoria V. Volchek, Nikolay B. Kompankov, Maxim N. Sokolov, Pavel A. Abramov

**Affiliations:** 1Nikolaev Institute of Inorganic Chemistry SB RAS, 3 Akad. Lavrentiev Ave., 630090 Novosibirsk, Russia; 2Institute of Natural Sciences and Mathematics, Ural Federal University Named after B.N. Yeltsin, 620075 Ekaterinburg, Russia

**Keywords:** proton transfer, octamolybdate, NMR, chromatography, structural analysis

## Abstract

The affinity of [β-Mo_8_O_26_]^4−^ toward different proton sources has been studied in various conditions. The proposed sites for proton coordination were highlighted with single crystal X-ray diffraction (SCXRD) analysis of (Bu_4_N)_3_[β-{Ag(py-NH_2_)Mo_8_O_26_]}] (**1**) and from analysis of reported structures. Structural rearrangement of [β-Mo_8_O_26_]^4−^ as a direct response to protonation was studied in solution with ^95^Mo NMR and HPLC-ICP-AES techniques. A new type of proton transfer reaction between (Bu_4_N)_4_[β-Mo_8_O_26_] and (Bu_4_N)_4_H_2_[V_10_O_28_] in DMSO results in both polyoxometalates transformation into [V_2_Mo_4_O_19_]^4−^, which was confirmed by the ^95^Mo, ^51^V NMR and HPLC-ICP-AES techniques. The same type of reaction with [H_4_SiW_12_O_40_] in DMSO leads to metal redistribution with formation of [W_2_Mo_4_O_19_]^2−^.

## 1. Introduction

Protons play a key role in a wide range of water-associated processes, from geochemistry to biology [1]. The appearance of the Theory of Coupled Electron and Proton Transfer Reactions [2,3,4] opened rich prospects for chemical reactions design [5,6,7,8]. Currently, proton coupled electron transfer (PCET) processes play crucial roles in synthesis and catalysis [9,10], e.g., artificial photosynthesis systems [11,12,13,14] and PCET at interfaces [15,16,17,18].

In polyoxometalate (POM) chemistry, protonation affects the formation, stability and reactivity of polyoxoanions. Most self-assembly cascade reactions are pH driven when fast protonation-deprotonation processes provoke rapid species transformation/organization into various associates up to nanoscopic size. The study of self-assembly processes is one of the top subjects in modern chemical science [19,20,21,22,23,24,25,26,27]. Such a specific organization of the matter in different solutions is a research focus for a large number of research groups. For example, research groups led by T. Mak and Di Sun successfully merged polyoxometalate chemistry with that of coinage metal clusters using the self-assembly approach [28,29,30].

The electronic structure of polyoxoanions together with low-energy protonation makes such objects very attractive for PCET reactions. The most important catalytic process in this field is water oxidation [31,32]. Such POM catalysts as [{Ru^IV^_4_(OH)_2_(H_2_O)_4_}(γ-SiW_10_O_34_)_2_]^10−^ [33,34,35] and [Co^II^_4_(H_2_O)_2_(B-α-PW_9_O_34_)_2_]^10−^ have become classics [36,37]. Recently, [V_6_O_13_(TRIOL^NO^_2_)_2_]^2−^ was applied to achieve concerted transfer of protons and electrons. Fully reduced clusters can induce 2e^−^/2H^+^ transfer reactions from surface hydroxide ligands [38].

In the chemistry of group 6 polyoxometalates, the polyoxomolybdates are significantly more labile than the polyoxotungstates, thus making researchers favor the latter in their studies of POM chemistry. However, several studies of polyoxomolybdates’ reactivity [39] and catalytic performance (electron transfer reactions) appeared [40,41,42,43,44,45]. One of the central complexes in this chemistry is (Bu_4_N)_4_[β-Mo_8_O_26_] (Figure 1), which is a standard precursor of all reactions in organic media, leading to a huge number of materials with different properties [46,47,48,49]. Our ongoing research focuses on the use of the coordination chemistry of the [β-Mo_8_O_26_]^4−^ anion in the study of silver chemistry in non-aqueous solutions [50,51,52]. Karoui and Ritchie used (Bu_4_N)_4_[β-Mo_8_O_26_] in the microwave-assisted synthesis of tris(alkoxo)molybdovanadates [V_3_Mo_3_O_16_(O_3_-R)]^2−^ (R = C_5_H_8_OH or C_4_H_6_NH_2_) by the reaction between [β-Mo_8_O_24_]^4−^, [H_3_V_10_O_28_]^3−^ and pentaerythritol or tris(hydroxymethyl)aminomethane [53]. These results show the possibility of the reaction between two different types of polyoxometalates producing mixed-metal compounds based on a different structural type. Such reactions are practically unknown and can generate interesting mixed metal complexes. This is very important and can be used for various materials preparation applied in catalysis (different Mo/V oxides), photochemistry, solid-state devices (capacitors), biochemistry and biomedicine.

An important question is what is the trigger and the driving force of such metal redistribution reactions? In this research, we focused on the behavior of the [β-Mo_8_O_26_]^4−^ anion toward protonation to answer this question. Some years ago, we suggested a straightforward hyphenated HPCL-ICP-AES technique [54] as an efficient tool to study the reaction products in different polyoxometalate systems [55,56,57]. In the present research, this technique helps us to have control over products’ formation in different conditions.

## 2. Results

### 2.1. Structural Analysis

The structure of [β-Mo_8_O_26_]^4−^ is preorganized for the coordination of different metal cations due to the presence of two trans-located lacunes (Figure 1). During the study of complexation in the Ag^+^/[β-Mo_8_O_26_]^4−^/L (L = auxiliary ligand) systems [50,51,58], we found a large number of equilibria that can be easily shifted by the addition of different ligands. In the present case, we tested 4-aminopyridine (py-NH_2_) as an auxiliary ligand in order to produce a 1D {-Mo_8_-Ag-py-NH_2_-Ag-Mo_8_-} coordination polymer. Instead of this, the reaction gives (Bu_4_N)_3_[Ag(py-NH_2_)Mo_8_O_26_] as the main product (phase purity was confirmed by XRPD, see Appendix A). In the crystal structure (SCXRD details are collected in Appendix A) Ag^+^, [β-Mo_8_O_26_]^4−^ and py-NH_2_ combine into another type of 1D coordination polymer when [Ag(py-NH_2_)Mo_8_O_26_]^3−^ anions stack together via py-NH_2_…O=Mo interactions (Figure 1).

Three typical bonding distances surround Ag^+^: d(Ag1-N1) = 2.27(3) Å, d(Ag1-O6) = 2.372(7) Å, d(Ag1-O2) = 2.544(6) Å and two longer contacts d(Ag1-O9) = 2.639(7) and d(Ag1-O13) = 2.689(7) Å indicate CN = 3+2 for Ag^+^. These distances are in agreement with the previously published pyridinium complexes of this type [50]. The distances for py-NH_2_…O=Mo interactions fill the interval between 2.974 and 3.285 Å. The shortest N…O contacts 2.974 and 3.021 Å are depicted in blue in Figure 1.

The formation of this coordination polymer via NH_2_…POM interactions is very interesting. We did the structural search for bonding between the oxoligands of the [β-Mo_8_O_26_]^4−^ lacunes and H-atoms, and collected 11 hits (BURBOH, CASNIU, COPFIW, EWILIG, GEBYER, GISHEW, HIJSUR, MAXPUZ, MEPNIH, VEHTAF, YAGNOJ) from CCDC (ConQuest Version 2020.2.0). We will use the corresponding refcodes of deposited crystal structures as references in the description below.

The interactions between [β-Mo_8_O_26_]^4−^ and Me_2_NH(R), Me_2_NH_2_^+^ and NH_4_^+^ in the crystal structures of YAGNOJ (a); BURBOH (b); HIJSUR (c); GISHEW (d) are shown in Figure 2.

According to the structural analysis, R_3_NH^+^, R_2_NH_2_^+^ and NH_4_^+^ interact with terminal O=Mo groups of [β-Mo_8_O_26_]^4−^ lacunes. Moreover, even Me_4_N^+^ can interact with the lacune (VEHTAF). In the crystal structure of GEBYER, the [β-Mo_8_O_26_]^4−^ lacunes interact with two H_2_O molecules. In the case of **1**, we detected interaction between the neutral NH_2_-group protons with the O=Mo groups of polyoxomolybdate. This illustrates strong attraction between the lacune terminal oxoligands and H-atoms possessing some acidity (chiefly N–H, but also C-H in Me_4_N^+^). Considering this, we can suggest direct proton transfer exactly to these oxoligands-producing terminal Mo-OH group, which is highly reactive (M–O π-bonding breaking) and initiates further rearrangement of octamolybdate into hexamolybdate. The detailed mechanistic studies of this transformation are still absent. In this research, we used this channel to initiate the reaction between [β-Mo_8_O_26_]^4−^ and different protonated polyoxometalates serving as proton source. Such direct reactions between two different polyoxometalates are poorly studied. The HPLC-ACP-AES technique was used to control the products.

### 2.2. Reactivity of [β-Mo_8_O_26_]^4−^

The first candidate for this type of reaction was easily prepared (Bu_4_N)_4_H_2_[V_10_O_28_]. The HPLC-ICP-AES chromatogram of pure (Bu_4_N)_4_[β-Mo_8_O_26_] in acetonitrile shows a major molybdenum peak (t_R_ = 3.6 min), corresponding to the octamolybdate anion [β-Mo_8_O_26_]^4−^, and a minor peak (t_R_ = 4.8 min), which can be assigned as a hexamolybdate anion [Mo_6_O_19_]^2−^ (Figure 3a) [59]. The profile of the major peak is asymmetric due to the presence of [α-Mo_8_O_26_]^4−^, according to the previous ESI-MS data, demonstrating the absence of any other molybdates in the solution [50]. The HPLC-ICP-AES chromatogram of a freshly prepared solution of (Bu_4_N)_4_H_2_[V_10_O_28_] shows a single peak containing vanadium (t_R_ = 3.0 min), which confirms the presence of individual vanadate anion [V_10_O_28_]^6−^ in the solution (Figure 3b). Moreover, the addition of 2 eq of Bu_4_NOH to the solution of (Bu_4_N)_4_H_2_[V_10_O_28_] does not reflect any POM transformation.

The reaction between (Bu_4_N)_4_[β-Mo_8_O_26_] and (Bu_4_N)_4_H_2_[V_10_O_28_] in DMSO does not proceed at room temperature, according to ^51^V NMR data, which is the fastest way to check the reaction progress. The reaction mixture must be heated over 50 °C to activate the polyoxometalates’ transformation. The HPLC-ICP-AES technique was used to investigate the reaction products at different molar ratios of the reagents. The reaction time was 10 min.

For molar ratio 5/1 (Mo:V = 5/1) at C_o_ of (Bu_4_N)_4_[β-Mo_8_O_26_] = 6 mM, we observed one peak with the atomic ratio Mo:V = 2.2 (t_R_ = 4.3 min) (Figure 4a) and a second V-free peak (t_R_ = 4.8 min), which may be ascribed to unreacted octamolybdate (Figure 4a). With an increase in the vanadate concentration (Mo/V = 5:2 molar ratio), the same major peak with atomic ratio Mo:V = 2.5 was observed, the intensity of which doubled (Figure 4b). In addition, a chromatogram shows a minor Mo-free peak (t_R_ = 3.0 min), which indicates an excess of the decavanadate anion in this case (Figure 3b).

No significant changes in the chromatograms were observed with a further increase in the concentration of vanadate. The atomic ratio Mo:V = 2.2 indicates the formation of [V_2_Mo_4_O_19_]^4−^ Lindqvist type anions as the reaction product. According to ^51^V NMR, the total intensity of the other V peaks is ca. 1% of the intensity of the signal from the major product (See NMR part).

The next candidate to study the proton transfer controlled reaction with (Bu_4_N)_4_[β-Mo_8_O_26_] was [H_4_SiW_12_O_40_]·14H_2_O. Preliminary experiments showed that the reaction between (Bu_4_N)_4_[β-Mo_8_O_26_] and [H_4_SiW_12_O_40_]·14H_2_O in acetonitrile proceeded slowly and led to the formation of a number of products in comparable amounts. Therefore, CH_3_CN was replaced with dimethyl sulfoxide (DMSO). The HPLC-ICP-AES chromatogram of a freshly prepared solution of silicotungstic acid in DMSO shows a single peak containing tungsten (t_R_ = 6.2 min), which confirms the presence of individual silicotungstate anion [SiW_12_O_40_]^4−^ in the solution (Figure 5a) (The intensities of Si lines are significantly lower than W or Mo and cannot be adequately estimated).

The HPLC-ICP-AES chromatogram of (Bu_4_N)_4_[β-Mo_8_O_26_] in DMSO (Figure 5b) is similar to the chromatogram in acetonitrile (Figure 3a) and shows a major molybdenum peak (t_R_ = 4.5 min), corresponding to the octamolybdate anion [β-Mo_8_O_26_]^4−^, and a minor peak of [Mo_6_O_19_]^2−^ [59]. Since the viscosity of DMSO is 5 times that of acetonitrile, we were forced to reduce the concentration of the ion-pair reagent in the HPLC eluent to prevent column overpressure. Therefore, the peak retention times in DMSO increased. The HPLC-ICP-AES technique was used to investigate the reaction products between (Bu_4_N)_4_[β-Mo_8_O_26_] and [H_4_SiW_12_O_40_]·14H_2_O at different molar ratios. For the Mo/W = 10/1 molar ratio at C_o_ of (Bu_4_N)_4_[β-Mo_8_O_26_] = 3 mM, we observed four peaks (Figure 6a): (i) unreacted octamolybdate (t_R_ = 4.5 min), (ii) poorly separated peak with atomic ratio Mo:W = 2.3 (t_R_ = 4.7 min), (iii) hexamolybdate (t_R_ = 5.7 min) and (iv) Mo-free peak (t_R_ = 6.2 min) from unreacted silicotungstic acid. With an increase in the tungstate concentration (Mo/W = 10/2 molar ratio), the same major peak with atomic ratio Mo:W = 2.3 was observed (Figure 6b). In addition, the chromatogram shows minor W-free peaks (t_R_ = 4.5 min, t_R_ = 5.6 min) and a single peak containing tungsten (t_R_ = 6.2 min), which may indicate an excess of the tungstate anion. Further increase in the concentration of tungstate (Mo/W = 10/4 molar ratio) leads to the disappearance of the first molybdenum peak ([β-Mo_8_O_26_]^4−^, t_R_ = 4.5 min) and an increase in the intensity of the peak of unreacted tungstate.

Thus, according to the HPLC-ICP-AES results, the proton transfer between the silicotungstic acid and [β-Mo_8_O_26_]^4−^ triggers metal redistribution with the formation of Lindqvist type [W_2_Mo_4_O_19_]^2−^ anion as the reorganization product of [β-Mo_8_O_26_]^4−^. Curiously, [α-Mo_8_O_26_]^4−^ does not react in this case. The Keggin anion almost completely converts into the mixed Lindqvist at Mo/W ratio = 10/1 (Figure 6a). We reported a similar process earlier, when direct reaction of [H_3_PW_12_O_40_] with [NbO(C_2_O_4_)_2_]^−^ yielded [PW_11_NbO_40_]^4−^ [60].

The reaction between (Bu_4_N)_4_[β-Mo_8_O_26_] and acetic acid was investigated with the HPLC technique. The reaction was run in DMSO (C_o_ of = 3 mM) by the addition of various concentrations of acetic acid (Figure 7).

The HPLC chromatogram of a freshly prepared solution of (Bu_4_N)_4_[β-Mo_8_O_26_] shows the peaks from octamolybdate [β-Mo_8_O_26_]^4−^ (Figure 5, peak no. 3, t_R_ = 4.5 min) and hexamolybdate [Mo_6_O_19_]^2−^ (Figure 7, peak no. 5, t_R_ = 5.6 min) in the ratio of 95:5. Addition of 0.001 M acetic acid decreases the octamolybdate peak intensity, while causing an increase in the hexamolybdate peak intensity and the appearance of a new peak (peak no. 4, t_R_ = 5.1 min). Further increase in the acetic acid concentration continues to reduce the intensity of the octamolybdate peak and leads to an increase in the intensity of peak no. 4, as well as the appearance of two minor peaks (peak no. 1,2) of smaller molybdates. At an acetic acid concentration of 0.008 M, the intensity ratio of the peaks corresponding to octamolybdate (peak no. 3), the new product (peak no. 4), and hexamolydate (peak no. 5), is 1.8:3.4:1, respectively. No further changes in the ratio of species in solution was observed with an increase in the concentration of acetic acid from 0.008 M to 0.01 M; however, the intensity of all peaks decreases by 1.5, and further acidification leads to the formation of a white precipitate, which makes the HPLC analysis unapplicable.

From this observation it follows that the transformation of [β-Mo_8_O_26_]^4−^ into [Mo_6_O_19_]^2−^ can be explained as direct dimolybdate ([Mo_2_O_7_]^2−^) elimination, as was proposed earlier. There are two simple molybdate anions in the reaction mixture. In the literature there is a structure of K[MoO_2_(OAc)_3_]·HOAc [61] complex, showing the possibility of [MoO_2_(OAc)_3_]^−^ existence in the solution. The new peak (Figure 7, peak no. 4) can be assigned as [Mo_8_O_24_(OAc)_2_]^4−^, with the same structure as reported for the malonate derivative ((NH_4_)_4_[Mo_8_O_24_(C_3_H_2_O_2_)_2_]·4H_2_O) [62].

### 2.3. NMR

NMR spectroscopy was anticipated to be an informative tool to study the reaction between (Bu_4_N)_4_[β-Mo_8_O_26_] (**Mo8**) and (Bu_4_N)_4_H_2_[V_10_O_28_] (**V10**) due to the presence of both ^51^V and ^95^Mo NMR active isotopes. We measured ^95^Mo NMR spectra for the following solutions to study the effects of acidification of **Mo8** by Hpts (Hpts = *p*-toluenesulfonic acid) (Figure 8).

As can be seen, addition of Hpts as a non-coordinating organic acid to the solution of (Bu_4_N)_4_[β-Mo_8_O_26_] leads to the disappearance of the [α-Mo_8_O_26_]^4−^ isomer and an increase in the amount of [Mo_6_O_19_]^2−^. Simple (mononuclear or binuclear) Mo-containing complexes were not detected, most likely due to the fast exchange. The signals from such species should appear in a negative region, in comparison with the literature [63].

The reaction between **Mo8** and **V10** was studied using both ^95^Mo and ^51^V NMR (Figure 9).

The ^51^V NMR spectra (Appendix A) show exclusive formation of [V_2_Mo_4_O_19_]^4−^, to the detriment of other mixed metal Lindqvist molybdovanadates, meaning that such reactions can offer a straightforward way to this anion. Two signals in the ^95^Mo NMR spectrum of (Bu_4_N)_4_[β-Mo_8_O_26_] indicate an equilibrium between α and β isomers, as described in the literature [64]. In the case of spectrum *b* (Figure 9), the baseline correction was not as accurate, and the peaks from **Mo8** have slightly negative intensities. Moreover, due to this problem, the profile of the main signal is also not as correct. Nevertheless, we can postulate the presence of [V_2_Mo_4_O_19_]^4−^ and [Mo_6_O_19_]^2−^ in the reaction mixture.

## 3. Materials and Methods

### 3.1. Physical Methods

(Bu_4_N)_4_[β-Mo_8_O_26_], (Bu_4_N)_2_[Mo_6_O_19_], (Bu_4_N)_2_[Mo_2_O_7_], (Bu_4_N)_4_H_2_[V_10_O_28_] and (Bu_4_N)_3_Na[V_2_Mo_4_O_19_] were prepared according to the literature data (Inorg. Synth. Vol. 27). DMSO was distilled in vacuo over NaOH. [H_4_SiW_12_O_40_]·14H_2_O was manufactured by “The Red Chemist” (Saint Petersburg, USSR) and checked with FT-IR and TGA prior to use. Other reagents were of commercial quality (Sigma Aldrich) and were used as purchased. IR spectra were recorded on a Bruker Vertex 60 FT-IR spectrometer. Elemental analysis was carried out on a MICRO Cube CHN analyzer.

**Synthesis of (Bu_4_N)_3_[Ag(py-NH_2_)Mo_8_O_26_] (1)**: (Bu_4_N)_4_[β-Mo_8_O_26_] (200 mg, mmol) was dissolved in 2 mL of DMF under sonication, and afterward, 20 mg (mmol) of py-NH_2_ was added to the observed clear solution. Solid AgNO_3_ (32 mg, mmol) was added to the reaction mixture under sonication. The resulting mixture was placed in Et_2_O atmosphere at 4 °C to obtain crystalline material. The crop of large colorless crystals was isolated after 48 h. Yield was 175 mg (90% based on initial octamolybdate).

Elemental analysis. Calcd C, H, N (%) for **1**: 30.1, 5.4, 3.3; found C, H, N (%): 30.0, 5.3, 3.2.

IR (KBr, cm^−1^): 3485 (w), 3336 (m), 3222 (w), 2959 (vs), 2929 (s), 2872 (s), 1632 (vs), 1611 (s), 1554 (w), 1522 (m), 1480 (vs), 1456 (s), 1375 (m), 1355 (w), 1333 (w), 1278 (w), 1214 (m), 1148 (w), 1100 (w), 1060 (w), 1028 (w), 1014 (m), 970 (s), 947 (vs), 928 (vs), 905 (vs), 865 (vs), 847 (vs), 825 (s), 811 (m), 700 (vs), 657 (vs), 567 (m), 551 (s), 520 (s), 472 (m), 444 (w), 407 (s).

### 3.2. NMR

^51^V and ^95^Mo NMR spectra were recorded on a Bruker Avance III 500 spectrometer (BBI detector), using NaVO_3_ and Na_2_MoO_4_ as internal standards. Spectra were measured in DMSO-*d*_6_ at room temperature using standard 5 mm NMR tubes.

### 3.3. X-ray Diffraction on Single Crystals

Crystallographic data and refinement details are given in Appendix A). The diffraction data for **1** were collected on a Bruker D8 Venture diffractometer with a CMOS PHOTON III detector and IµS 3.0 source (Mo Kα radiation, λ = 0.71073 Å) at 150 K. The φ- and ω-scan techniques were employed. Absorption correction was applied by SADABS (Bruker Apex3 software suite: Apex3, SADABS-2016/2 and SAINT, version 2018.7-2; Bruker AXS Inc.: Madison, WI, USA, 2017). Structures were solved by SHELXT [65] and refined by full-matrix least-squares treatment against |F|^2^ in anisotropic approximation with SHELX 2014/7 [66] in the ShelXle program [67]. H-atoms were refined in geometrically calculated positions.

CCDC 2215910 contains the supplementary crystallographic data. These data can be obtained free of charge via http://www.ccdc.cam.ac.uk/conts/retrieving.html, or from the Cambridge Crystallographic Data Centre, 12 Union Road, Cambridge CB2 1EZ, UK; fax: (+44) 1223-336-033; or e-mail: deposit@ccdc.cam.ac.uk.

### 3.4. XRPD

X-ray powder diffraction patterns were measured on a Bruker D8 Advance diffractometer using LynxEye XE T-discriminated CuKα radiation. Samples were layered on a flat plastic specimen holder.

### 3.5. HPLC-ICP-AES and HPLC

Separation was performed with the HPLC system Milichrom A-02 (EcoNova, Novosibirsk, Russia), equipped with a two-beam spectrophotometric detector at the wavelength range of 190−360 nm in the ion-pair mode of reversed phase chromatography (ProntoSIL 120-5-C18AQ, 2 × 75 mm), eluents: A—0.06% tetrabutylammonium hydroxide (for (Bu_4_N)_4_[β-Mo_8_O_26_] and (Bu_4_N)_4_H_2_[V_10_O_28_]), 0.02% tetrabutylammonium hydroxide (for (Bu_4_N)_4_[β-Mo_8_O_26_] and [H_4_SiW_12_O_40_]·14H_2_O); B—acetonitrile. Gradient elution with a gradual increase in acetonitrile concentration was employed to resolve the species. ICP-AES spectrometer iCap 6500 Duo (Thermo Scientific, Waltham, MA, USA) with concentric nebulizer was applied as detector in hyphenated HPLC-ICP-AES. For the element detection Mo 281.6 nm, V 292.4 nm and W 239.7 nm, spectral lines were selected. All measurements were performed in three replicates.

The data acquisition and processing were carried out with iTEVA (Thermo Scientific, Waltham, MA, USA) software. In order to eliminate plasma quenching, we diluted the liquid coming out of the column into the spray chamber with deionized water. The steady state of the plasma and the optimal values of the analytical signals were finally achieved at the eluent flow rate of 0.25 mL min^−1^ and the eluent velocity of 3 mL min^−1^ (peristaltic pump speed—75 rpm).

## 4. Conclusions

This manuscript describes an affinity of [β-Mo_8_O_26_]^4−^ lacunes for interaction with H-atoms possessing some N–H or even C–H (in Me_4_N^+^) acidity. We demonstrated this in the case of 1D polymeric chains formation via py-NH_2_ and [β-Mo_8_O_26_]^4−^ interaction. This example illustrates a general approach to the formation of soft matters based on such types of interactions. The reaction of [β-Mo_8_O_26_]^4−^ with diluted acids generates a set of unknown complexes, according to the HPLC-ICP-AES data. Moreover, there was a new type of reactivity of [β-Mo_8_O_26_]^4−^ combining: (i) proton transfer from another type of polyoxometalates in solution, (ii) backbone breaking and (iii) transformation into mixed Lindqvist type complexes has been demonstrated. In the case of (Bu_4_N)_4_H_2_[V_10_O_28_], this reaction gives [V_2_Mo_4_O_19_]^4−^. [H_4_SiW_12_O_40_] plays a role as a proton and W source, producing [W_2_Mo_4_O_19_]^2−^. The key study here is proton transfer into the lacune of [β-Mo_8_O_26_]^4−^, generating the reactive transition state. At the current stage, it is impossible to deduce the mechanism, which is not as simple as [Mo_2_O_7_]^2−^-elimination. In comparison with the microwave synthesis reported by Karoui and Ritchie, simple thermal activation does not need any special equipment. The addition of any triol type organic ligands into the reaction mixture will be the next step in such reactivity studies. Such an approach opens a way to new mixed functionalized complexes.

## Data Availability

The crystallographic data have been deposited in the Cambridge Crystallographic Data Centre under the deposition codes CCDC 2215910.

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
