# Peer review of "Proton Affinity in the Chemistry of Beta-Octamolybdate: HPLC-ICP-AES, NMR and Structural Studies"

_molecules, 2022, doi:10.3390/molecules27238368_

Round 1

Reviewer 1 Report

The manuscript by Abramov and coworkers report on a new SC X-Ray structure between octamolybdate and 4-aminopyridine coordinated silver. The authors propose a proton coordination site based on their structure comparing it to previously reported structures in literature. Proton transfer between octamolybdate and decavanadate/silicotungstate was also investigated to provided a better understanding of the role of protonation in metal redistribution reactions, leading to a simpler method to generate the mixed metal V2Mo4 and W2Mo4 Lindqvist ions compared to the microwave assisted synthesis reported by Karoui and Ritchie to isolated the V3Mo3-R hybrid counterpart. Although the science is sound, the research does not have a large impact or a broad audience.

The two parts (XRD and HPLC studies) within the paper seem detached and the link between them is not quite clear (the manuscript needs rearrangement to make the research question more clear for readers and a better connection between the two parts is needed).

The authors should ask a native English speaker to proofread the paper before publication especially when it comes to articles and grammar.

Some inconsistencies in the manuscript (the use of polymolybdate, polyoxomolybdate, polyoxotungstate, polyoxometalate...). 

Line 44: However, several studies the studies (repetition)...

Line 114: the authors assigned the peaks at tR 3.6 min to Mo8 octamolybdate and that at tR 4.8 min to Mo6 Lindqvist (in CH3CN) while later in the text (line 128) they mention that the peak at tR 4.8 min correspond to unreacted octamolybdate in (DMSO) without mentioning this until later in the text (line 155) where the chromatogram of Mo8 in DMSO is presented which raises confusion before reaching this section!

Section 2.3.: NMR lines 215-217 all 3 experiments say fig 8a this doesn't match with the figure

Figure 9 b (the two peaks withing broad profile at 129 ppm should be assigned different chemical shifts to match the author's description indicating it corresponds to V2Mo4 and Mo6 Lindqvists) 

The conclusions are supported by the experiments and address the research question.

Author Response

The manuscript by Abramov and coworkers report on a new SC X-Ray structure between octamolybdate and 4-aminopyridine coordinated silver. The authors propose a proton coordination site based on their structure comparing it to previously reported structures in literature. Proton transfer between octamolybdate and decavanadate/silicotungstate was also investigated to provided a better understanding of the role of protonation in metal redistribution reactions, leading to a simpler method to generate the mixed metal V2Mo4 and W2Mo4 Lindqvist ions compared to the microwave assisted synthesis reported by Karoui and Ritchie to isolated the V3Mo3-R hybrid counterpart. Although the science is sound, the research does not have a large impact or a broad audience.

The presented study combines two very important research dimensions:

  1. i) Formation of new type soft-matter materials based on octamolybdate and N-H acidity bearing building blocks. Such objects can be e.g. redox sensors or photocatalysts dependently on organic ligand;
  2. ii) New type reaction in polyoxometalate chemistry based on interaction between two different polyoxometalates via proton transfer. This reaction can generate mixed metal complexes for interesting catalytic applications including photocatalysis and so on.

Both of these points are of highly importance in modern coordination and inorganic chemistry.

The two parts (XRD and HPLC studies) within the paper seem detached and the link between them is not quite clear (the manuscript needs rearrangement to make the research question more clear for readers and a better connection between the two parts is needed).

The main text has been updated to reach a better connection between the two parts.

The authors should ask a native English speaker to proofread the paper before publication especially when it comes to articles and grammar.

A native English speaker checked the manuscript.

Some inconsistencies in the manuscript (the use of polymolybdate, polyoxomolybdate, polyoxotungstate, polyoxometalate...). 

This was fixed!

Line 44: However, several studies the studies (repetition)...

Thank you very much! This was fixed!

Line 114: the authors assigned the peaks at tR 3.6 min to Mo8 octamolybdate and that at tR 4.8 min to Mo6 Lindqvist (in CH3CN) while later in the text (line 128) they mention that the peak at tR 4.8 min correspond to unreacted octamolybdate in (DMSO) without mentioning this until later in the text (line 155) where the chromatogram of Mo8 in DMSO is presented which raises confusion before reaching this section!

Octamolybdate and hexamolybdate have different retention times in CH3CN and DMSO. In CH3CN octamolybdate has tR 3.6 min and hexamolybdate has tR 4.8 min. In DMSO these values are 4.5 and 5.7 min correspondingly. Since the viscosity of DMSO is 5 times that of acetonitrile, we were forced to reduce the concentration of the ion-pair reagent in the HPLC eluent to prevent column overpressure. Therefore, the peak retention times in DMSO increased. The mentioned part in the main text has been updated.

Section 2.3.: NMR lines 215-217 all 3 experiments say fig 8a this doesn't match with the figure

Thank you very much! This was fixed!

Figure 9 b (the two peaks withing broad profile at 129 ppm should be assigned different chemical shifts to match the author's description indicating it corresponds to V2Mo4 and Mo6 Lindqvists) 

Fig. 9 has been modified!

The conclusions are supported by the experiments and address the research question.

Reviewer 2 Report

In this study, the authors described synthesis process and crystal structure of a beta-octamolybdate based hybrid (Bu4N)3[β-{Ag(py-NH2)Mo8O26]}] (1). Also, the proton transfer properties of (Bu4N)4[β-Mo8O26] and (Bu4N)3[H3V10O28] were studied. The (Bu4N)4[β-Mo8O26] and (Bu4N)3[H3V10O28] in DMSO results in polyoxomolybdate transformation into a set of mixed V/Mo Lindqvist species. In general, the article is interesting and can be reconsidered for publication after a major revision.

1.     Abstract must be enriched via valuable results which pave the way for understanding the audiences.

2.     The proton transfer properties of(Bu4N)4[β-Mo8O26] and (Bu4N)3[H3V10O28] were studied. However, the authors only described synthesis process and crystal structure of a new beta-octamolybdate based hybrid (Bu4N)3[β-{Ag(py-NH2)Mo8O26]}]. How about the proton transfer properties of the compound (Bu4N)3[β-{Ag(py-NH2)Mo8O26]}]?

3.     Why does the research focus on beta-octamolybdate based hybrid?

4.     Indeed, there are impressive amounts of results. However, the conclusions section needs to improve with selected and highlighted main findings.

Author Response

In this study, the authors described synthesis process and crystal structure of a beta-octamolybdate based hybrid (Bu4N)3[β-{Ag(py-NH2)Mo8O26]}] (1). Also, the proton transfer properties of (Bu4N)4[β-Mo8O26] and (Bu4N)3[H3V10O28] were studied. The (Bu4N)4[β-Mo8O26] and (Bu4N)3[H3V10O28] in DMSO results in polyoxomolybdate transformation into a set of mixed V/Mo Lindqvist species. In general, the article is interesting and can be reconsidered for publication after a major revision.

  1. Abstract must be enriched via valuable results which pave the way for understanding the audiences.

Thank you very much! The abstract has been updated.

  1. The proton transfer properties of (Bu4N)4[β-Mo8O26] and (Bu4N)3[H3V10O28] were studied. However, the authors only described synthesis process and crystal structure of a new beta-octamolybdate based hybrid (Bu4N)3[β-{Ag(py-NH2)Mo8O26]}]. How about the proton transfer properties of the compound (Bu4N)3[β-{Ag(py-NH2)Mo8O26]}]?

The main idea of the MS is illustration of a new type reaction based on proton transfer from one POM to another with transformation of both precursors. (Bu4N)3[β-{Ag(py-NH2)Mo8O26]}] was included into the manuscript to illustrate the most probable proton transfer route during the octamolybdate transformation. We used HPLC-ACP-AES and NMR techniques to control the reaction process.

The deprotonation of terminal NH2-group needs agents that are more basic. [β-Mo8O26]4– is not too strong for this point of view.

  1. Why does the research focus on beta-octamolybdate based hybrid?

(Bu4N)3[β-{Ag(py-NH2)Mo8O26]}] was included into the manuscript to illustrate the most probable proton transfer route during the octamolybdate transformation. This was a starting point before discovering of proton transfer triggered reactions.

  1. Indeed, there are impressive amounts of results. However, the conclusions section needs to improve with selected and highlighted main findings.

Thank you very much! The conclusions part has been updated.

Reviewer 3 Report

Several matters posed in the paper need to be addressed:

1)    Abstract need improvement to highlight the research study of manuscript.

2)    Page 2, line 58-60, Introduction: A clearer explanation on the correlation of previous study might be helpful to relate to current study.

3)    All figures, especially chromatograms, should be of similar size. Eg. Figure 3 is larger than Figure 4, 5, 6; while Figure 6 is relatively small.  

4)    For NMR studies, variable temperature experiments might be useful to be included for the structural studies.

5)    Conclusion is not clear and does not summarize objective of research.

In summary, the authors should have their work carefully checked for English

expression before submitting a revised version and proof-reading might be helpful. I therefore suggest MINOR REVISION.

Author Response

Several matters posed in the paper need to be addressed:

1) Abstract need improvement to highlight the research study of manuscript.

The abstract has been updated!

2) Page 2, line 58-60, Introduction: A clearer explanation on the correlation of previous study might be helpful to relate to current study.

The Introduction part has been changed.

3) All figures, especially chromatograms, should be of similar size. Eg. Figure 3 is larger than Figure 4, 5, 6; while Figure 6 is relatively small.  

Thank you very much for the comment! We did our best in the revised version.

4) For NMR studies, variable temperature experiments might be useful to be included for the structural studies.

This is very interesting point. We are planning to organize such type experiments to estimate the proton migration barrier. These data are going to be huge and will be separately published.

5) Conclusion is not clear and does not summarize objective of research.

Thank you very much! The conclusions part has been updated.

In summary, the authors should have their work carefully checked for English expression before submitting a revised version and proof-reading might be helpful. I therefore suggest MINOR REVISION.

Thank you very much! A native English speaker checked the MS.

Round 2

Reviewer 2 Report

This is a revised manuscript from a paper suggested by previous reviewers. In my opinion the authors have carefully revised their paper and adequately dealt with all the issues raised in the initial comments of the editor and reviewers.